# Wearable Technologies in Field Hockey Competitions: A Scoping Review

**DOI:** 10.3390/s21155242

**Published:** 2021-08-03

**Authors:** Jolene Ziyuan Lim, Alexiaa Sim, Pui Wah Kong

**Affiliations:** Physical Education and Sports Science Academic Group, National Institute of Education, Nanyang Technological University, Singapore 637616, Singapore; nie20.lzj@e.ntu.edu.sg (J.Z.L.); nie184704@e.ntu.edu.sg (A.S.)

**Keywords:** wearable, field hockey, competition analysis

## Abstract

The aim of this review is to investigate the common wearable devices currently used in field hockey competitions, and to understand the hockey-specific parameters these devices measure. A systematic search was conducted by using three electronic databases and search terms that included field hockey, wearables, accelerometers, inertial sensors, global positioning system (GPS), heart rate monitors, load, performance analysis, player activity profiles, and competitions from the earliest record. The review included 39 studies that used wearable devices during competitions. GPS units were found to be the most common wearable in elite field hockey competitions, followed by heart rate monitors. Wearables in field hockey are mostly used to measure player activity profiles and physiological demands. Inconsistencies in sampling rates and performance bands make comparisons between studies challenging. Nonetheless, this review demonstrated that wearable devices are being used for various applications in field hockey. Researchers, engineers, coaches, and sport scientists can consider using GPS units of higher sampling rates, as well as including additional variables such as skin temperatures and injury associations, to provide a more thorough evaluation of players’ physical and physiological performances. Future work should include goalkeepers and non-elite players who are less studied in the current literature.

## 1. Introduction

Field hockey is an invasion game that is played in four quarters over 60 min. This is after a rule change in 2015, as the game was previously played in two 35-min halves. As a fast-paced, intermittent sport, the physical fitness of field hockey players is important to the individual and the overall team performance [1,2,3]. Wearable technologies in sports have been on the rise. Sports such as soccer, rugby, and Australian football are turning to devices that provide easier in-depth monitoring of a player’s fitness progress as compared to the traditional video analysis [4,5,6]. With the increasing demand of wearables in team sports, many devices such as heart rate monitors, global positioning system (GPS) devices, and accelerometers have been developed to assess the different physical and physiological demands of the players during training and competitions [5,7,8]. The advancement of technology has also allowed the integration of multiple devices into one single sensor, to enable a more thorough evaluation of a player’s progress during training and competitions [1,9].

### 1.1. Global Positioning System (GPS)

Global positioning system (GPS) (Figure 1a) is a navigation system that was originally developed for the military but has since been made available for commercial use. It is based on connections to satellites that provide information regarding location and time through a sampling frequency measured in Hertz (Hz). The communication between the satellite and the GPS units allows a player’s position in space and time to be triangulated [10,11]. In order to establish accurate connections and eliminate bias, GPS units require at least four satellites for accurate measurements, and accuracy increases when more satellites are involved [11,12]. In addition to the number of satellites needed for an accurate measurement, the geometrical positions of these satellites also affect the dilution of precision (DOP), which indicates the quality of position triangulation. This is known as horizontal dilution of precision (HDOP). A higher HDOP indicates fewer satellites, and also that the satellites are tightly spaced. In contrast, a lower HDOP indicates more satellites, which are evenly spaced. An HDOP of one is the ideal value as it demonstrates the greatest accuracy of triangulation [11]. Though GPS technology is less accurate than video analysis [13,14,15], it is still considered the most useful for monitoring sporting performances [14].

Commonly integrated into GPS units are tri-axial accelerometers, magnetometers, and gyroscopes [16]. Together, these components provide movement information in the different planes. Tri-axial accelerometers calculate acceleration in the medial-lateral, anterior-posterior, and vertical planes, usually at a sampling rate of 100 Hz [17]. The sum of acceleration in the different planes allows body load to be calculated [18] during training and competitions. The accelerations are then categorized into intensity bands or speed zones for analysis. The addition of a gyroscope allows angular velocity to be measured in the transverse plane, providing additional information about the tackling and collision movement of the athletes. As gyroscopes measure rotational velocity, studies have used them to detect non-vertical motion, which the authors here associate with collisions [19]. Together with the data from the accelerometer, the researchers were able to categorize the intensity of these collisions, which is useful for identifying potential injury risks [19]. Magnetometers use a magnetic field to detect the orientation of the device [17]. This provides information regarding the direction in which players are facing or moving [20]. The combination of these three sensors provides a comprehensive analysis of the sporting performance of the players.

### 1.2. Heart Rate Monitors

Compared to GPS units, heart rate monitors (Figure 1b) are more straightforward. Through the advancement of various technologies used to measure heart rate, wireless heart rate monitors were eventually developed. These monitors include a transmitter and a receiver which allow heart rate to be measured through the time between two successful R-waves on the ECG signal, also known as the R-R interval [21]. Heart rate monitors used in sports are usually strapped to the chest to record heart rate data as a means of monitoring internal load [17] during training and competitions. The monitors are attached to a chest belt that includes embedded electrodes. Though they can be uncomfortable, chest-strapped heart rate monitors have shown higher accuracy than the more comfortable, wrist-worn monitors [21,22], and are currently the most commonly used devices to objectively assess workload and exertion [23].

### 1.3. Wearable Technologies in Sport

Traditionally, player movements during competitions were tracked using hand notational analysis. While this eventually advanced to video analysis, the process remains time consuming and laborious [9]. Video analysis, while reliable, is also expensive and requires extensive calibration. In contrast, wearable devices are convenient and can track real-time exertion data, which is not possible when using video footage. Wearable devices do not require exhaustive setup, they offer reliable and valid measurements [12,24], they gather immediate and real-time feedback, and they provide physiological and physical information about athletes.

Wearable devices have been extensively used in team sports such as cricket, football, soccer, rugby, and running [18] to objectively collect performance data rather than depending on subjective methods that are deemed less accurate [17]. These devices provide a thorough measurement of players’ physical and physiological fitness in real time during training and competitions. GPS tracking has been used to examine positional demands during competitions, which in turn provides relevant data such as injury tracking [25] and running performances according to goal status (win or lose). A study by Buccheit et al. [26] analyzed 384 professional senior soccer players from 16 teams that participated in the 2011 Asia Cup and found that running performance was greater when both opposing teams were tied, and lower as the goal difference increased. Positional data such as total distance, relative distance, velocity, acceleration/deceleration, and Player Load reflect the demands of each position (forwards, midfielders, defenders, and goalkeepers) during competitions. For instance, Bradley and Noakes [27] found that high-intensity running decreased in the second half of a soccer game when teams had a goal difference of one or less than one. The forwards were also found to have undertaken 15% and 54% more high-intensity running and sprinting in matches that were won. In rugby [28], GPS data indicated that backs performed more high-intensity running, which may be associated with the increase in creatine kinase activity, though the study only compared creatine kinase levels across time (baseline, pre-, and post-match) rather than across positions (forwards and backs). Nonetheless, wearable technologies such as GPS trackers are beneficial and may prompt coaches to make adjustments to training programs.

Despite the imprecision of GPS units, they are widely used to monitor the movement of players [29], while heart rate monitors inform the coaches about levels of exertion [18] when players are in play. This allows coaches to monitor and influence decisions, such as whether to take a player out to prevent extreme fatigue or injuries when they are performing near maximal exertion. In addition, wearable devices are small, light, and portable, which makes them convenient for data collection. The data obtained from the wearable devices can provide coaches with additional information on how to properly design training programs which can eventually lead to better performance [9].

### 1.4. Research Gap

Considering the various usages of wearable devices, it would be interesting to have an overview of how these devices are being used in field hockey. Field hockey differs from other sports in that it allows unlimited substitutions between players and, therefore, the physical demands specific to field hockey are very different from other sports. These devices play an essential role in assessing how players cope with fatigue [30] and allowing teams to maximize their physical and technical outputs [31]. This has important implications, especially during competitions where player performances have a huge impact on game outcomes.

A review by Cummins et al. [18] acknowledged that GPS technology in field hockey has been limited compared to sports such as soccer and rugby. Indeed, their review on GPS technology in team sports found only five papers that focused on field hockey. Given the increasing popularity of GPS in team sports in the past decade, it is likely that such technology is now more commonly used in field hockey. Moreover, as sports continue to embrace the convenience of wearables, GPS technology may not be the only wearable used in the analysis of field hockey player performances. Other devices that are convenient, light, and do not interfere with player performance during competitions may be available for real-time or post-game analysis in field hockey. To the authors’ knowledge, no study has collectively examined all wearable technologies that field hockey players might wear during their competitions. Hence, the aim of this study is to review the various types of wearable technologies and the associated parameters used to measure competition performances in field hockey. Different devices measure different parameters and, therefore, examining which wearables are commonly used will provide an insight into which hockey-specific parameters are being placed at higher importance during competitions. This will also examine whether there are any lesser-known wearable devices that may be beneficial to the players and that warrant more attention.

## 2. Materials and Methods

### 2.1. Search Strategy

A systematic search was conducted in PubMed, Web of Science, and SPORTDiscus from the earliest record. The search strategy included a combination of keywords [All Fields]: ‘field hockey’ AND (wearable* OR technology*) OR (accelerometer*) OR (GPS OR global positioning system* OR global positioning satellite* OR global positioning unit* OR global positioning device*) OR (inertial senor* OR inertial measurement) OR (‘heart rate’ monitor OR HR monitor) OR ‘load’ OR (performance AND analysis OR ‘game analysis’ OR ‘motion analysis’ OR ‘time motion analysis’) OR (movement OR ‘player activity’ OR profiles OR patterns) AND (match* OR competition* OR tournament*) OR (‘game play’ OR ‘match play’).

### 2.2. Eligibility Criteria

Studies that measured field hockey player performances using wearables only during competitions were eligible. Wearable devices include (but are not limited to) GPS units, inertial motion units (IMU), accelerometers, heart rate monitors, and any other devices worn by the players during competitions. Both male and female field hockey players were included, with no restrictions on country, age, or competitive level. Analysis of games for all durations (one game or a few games, one season or a few seasons) were included. Studies that were included reported any physiological and/or biomechanical measurements made by wearable devices, including but not limited to distance, velocity/speed, heart rate, acceleration, deceleration, and load. Only full-text journal articles published in English were eligible for inclusion. Systematic reviews were excluded. Studies that measured player activity profiles during training or indoors were excluded. Studies that conducted small-sided or simulated game settings were also excluded. Search results were independently screened by two researchers against the eligibility criteria. Any disagreements were resolved by discussion.

### 2.3. Data Extraction

Data relating to participants’ sex and competitive level were extracted. Type of wearable and wearable characteristics such as brand, mode, sampling rate, parameters of match analysis (distance, speed, min of play), and work rate patterns (velocity/ locomotor activity zones, heart rate zones) were also extracted.

## 3. Results

The electronic databases search resulted in a total of 1131 studies (250 from PubMed, 496 from Web of Science, and 385 from SPORTDiscus). After removing duplicates, 364 articles remained for title and abstract screening. 286 articles were deemed irrelevant and 78 full-text articles were assessed for eligibility. 39 articles were included in this review. The PRISMA flowchart is shown in Figure 2.

### 3.1. Partcipant and Competition Characteristics

Participant and competition characteristics of the studies are shown in Table 1. Of the 39 studies included in this review, most studies were conducted on either only female (N =17) or only male (N = 20) field hockey players. Only one study included both sexes [32], and another did not mention the sex composition [33]. Five studies were conducted on youths and the rest (N = 34) were in the open category. The competitive levels of these studies included elite (N = 31), sub-elite (N = 8), and novice (N = 1) field hockey players. The competitive level of players was not mentioned in one study [33].

In addition, the competition formats of the studies were different, with 23 studies played in two 35-min halves and 14 studies played in four 15-min quarters. Competition format was not mentioned in three studies [34,35,36].

### 3.2. Types of Wearables

The majority of the studies used GPS units in their measurement of player performances (N = 32, Table 1). Among the thirteen studies that used heart rate monitors, four studies used heart rate monitors alone, seven studies used a combination of GPS and heart rate monitors, one study used the Zephyr BioHarness (an integrated device that includes accelerator and heart rate monitor) [1], and one study used wearable active transponders that send signals to base stations placed around the field hockey pitch [30].

#### 3.2.1. Brands

GPSports was the most common GPS brand (N = 15). The SPI Elite (N = 8), SPI Pro (N = 4), SPI ProX (N = 1), and SPI HPU (N = 2) models were used for match analysis. This was followed by the brand Catapult (N = 13), whose MinimaxX (N = 9), Minimax Team 2.5 (N = 1), and OptimEye S5 (N = 2) models were used. One study used a combination of MinimaxX and OptimEye S5. The remaining GPS brands were VS Sport (N = 3; models: VX350, VX110; model not mentioned in one study) and STATSports (N = 1; model: Apex). These are shown in Table 1.

With the exception of two studies that used the Zephyr BioHarness 3 and Firstbeat Technologies Oy, heart rate monitors from the remaining thirteen studies were from Polar (Table 1). Among the different models of Polar heart rate monitors, two studies used Polar Team, four studies used Polar Team 2, one study used Team Sport, two studies used T31, and one study used Vantage. The models of heart rate monitors were not mentioned in three studies. The remaining two studies used a CorTemp data logger to measure core temperature, or active transponders for local positioning monitoring.

#### 3.2.2. Sampling Rates

Sampling rates of the GPS units varied substantially, including 1 Hz (N = 3), 4 Hz (N = 1), 5 Hz (N = 11), 10 Hz (N = 14) and 15 Hz (N = 2). One study did not specify the sampling rate of GPS units (Table 1).

For heart rate measurements using Polar, two studies that used Polar Team 2 sampled at 1000 Hz, while all other studies did not mention the sampling rates (Figure 3). The heart rate monitor in the Zephyr BioHarness 3 had a sampling rate of 250 Hz, and the study that used active transponders collected data at 1000 Hz but had an individual player frequency of 62.5 Hz that transmitted signals to base stations around the pitch [30].

### 3.3. Purpose of Wearables

GPS units were used in 32 studies for the analysis of player activity profiles such as distance, speed, and time (Table 1). Comparisons were made between positions (N = 24; [1,2,3,30,31,33,34,37,38,39,40,41,42,43,44,45,46,47,48,49,50,51,52,53]), matches (N = 11; [8,36,41,44,47,50,51,54,55,56,57]), competition format (halves and quarters; (N = 10; [2,3,31,32,37,42,45,58,59]), age groups (N = 2; [51,52]), opponent rankings (N = 1; [39]), and competition level (international vs. national; (N = 1; [60])). Other analyses include association of activity profile with injuries (N = 1; [35]), torso demands (N = 1; [46]), different methods of analysis (N = 3; [61,62,63]), and effects of substitution (N = 1; [30]).

Heart rate was recorded using heart rate monitors in fifteen studies, nine of which compared positions [1,2,33,38,40,49,52,53], four compared competition formats [2,31,32,53], and two compared matches [8,57]. One study compared time spent above heart rate threshold among players [64]. A CorTemp data logger was used to measure thermoregulation of goalkeepers during game play in one study [32].

### 3.4. Parameters Measured by Wearables

Distance was measured in all studies that used GPS units (N = 32), and speed was the second most measured parameter (N = 22). The GPS units also measured acceleration (N = 9), deceleration (N = 7), and torso angles (N = 1). Time was measured in fourteen studies, and core and skin temperature was measured in one study. Lastly, heart rate was measured in fifteen studies. The parameters measured in each study are shown in Table 1.

### 3.5. Performance Bands

Out of the 32 studies that used GPS analyses, performance activity bands were mentioned for locomotor (N = 4), velocity (N = 21), estimated energy (N = 4), and acceleration/deceleration efforts (N = 7). Among the 21 studies that used velocity bands, 12 studies categorized these bands into levels (e.g., low, moderate, high). Seven studies used zones, with three studies having six zones, two studies having five zones, and two studies having four zones (Table 2). One study [48] categorized velocity into five bands, and in another velocity was categorized into faster and slower players [51]. Energy bands were mentioned in four studies, with one study categorizing into high and low power, while the other studies included band ranges from low to high/maximal. Acceleration and deceleration bands were mentioned in seven studies, two of which categorized acceleration into high and low, and four of which opted for low-high/very high. One study did not mention terms [60]. Performance activity band ranges are shown in Table 2.

Among the seven studies that used heart rate bands for analysis, heart rate was categorized into low to maximal intensity (N = 4) and zones (N = 2). One study [58] did not have any terms for the different thresholds. Heart rate bands are shown in Figure 3.

## 4. Discussion

This review aims to summarize the types of wearable technologies and the associated parameters used to measure competition performances in field hockey. Based on the results of the 39 included studies, it is evident that GPS technology is the most popular wearable in the analysis of field hockey competitions, followed by heart rate monitoring. GPSports and Catapult are the two leading GPS brands, while Polar is the prominent brand for heart rate monitors.

### 4.1. Sampling Rates

Although GPS technology is becoming more common in field hockey analysis, there is a range in the sampling rates being used for GPS recording which makes comparisons between studies challenging. In this review, studies used GPS units of sampling rates from 1 Hz to 15 Hz. GPS units in most studies (N = 14) were sampled at 10 Hz, and a considerable number of studies (N = 11) were sampled at 5 Hz.

Studies that have performed comparisons between sampling rates of GPS units have found higher accuracy in those of higher sampling rates [65,66], though sampling at 15 Hz was no more accurate than sampling at 10 Hz [67]. This indicates that 15 Hz GPS units have similar accuracy to 10 Hz GPS units, and that 10 Hz GPS units are sufficiently accurate in measuring performance indicators in field hockey. Despite this evidence, almost half of the studies that used GPS units in this review opted for sampling rates below 10 Hz (N = 15). The accuracy of the studies that used low sampling rates may be questioned. For example, three studies used 1 Hz GPS units [31,47,60] for categorizing and detecting high speed intensity and distances. While the limitation of low sampling rate was acknowledged in two of the studies [31,60], Sunderland and Edwards [47] argued that their GPS units were valid for examining the variables that they investigated.

Furthermore, two different studies employed SPI Elite GPS to measure the performance of elite field hockey players but found contrasting results [37,47]. In the study by Sunderland and Edwards [47], which sampled at 1 Hz, total distance covered was the greatest in fullbacks as compared to the other positions. However, the study by Morencos et al. [37], which sampled at a higher frequency of 10 Hz, found that forwards, not fullbacks, covered the greatest distance. While there are other factors that may have contributed to the discrepancy in distance measurements, the low sampling rate in Sunderland and Edwards [47] could have resulted in a less accurate measurement of distance. Therefore, comparisons of player performances between studies with different GPS sampling rates are difficult and should be conducted with caution. Future studies should also avoid using GPS with sampling rates lower than 10 Hz to ensure a higher level of accuracy in measurements.

### 4.2. Applications of Wearables

The results of this review revealed that wearables in field hockey are commonly used to measure player profile activities (via GPS) and physiological demands (via heart rate monitors). It is interesting to note that many different applications are possible based on similar data obtained from the wearable devices. For example, GPS data have been used to make comparisons between playing positions [31], halves and quarters of matches [45], age groups [51], competitive levels [39,60], as well as across elite field hockey matches [8,42].

Other than measuring activity profiles, GPS technology has also been used to assess players’ risk of injury. Kim et al. [35] has examined the association between player profile activities during competitions and non-contact ankle and knee injuries, and Warman et al. [46] has used GPS units to analyze postural data, specifically torso flexion and extension angles of elite male field hockey players during competitions. These applications demonstrate the variety of ways in which GPS technology has been used to monitor postural demands and injury-related parameters of field hockey players.

Heart rate data were also used in different investigations, either to make comparisons between age groups [52], or to calculate training loads for correlation analyses between heart rate and other variables, such as rate of perceived exertion [57,61]. The combination of GPS and heart rate monitors provided additional analyses on the physiological demands and loads experienced by the players during their games. GPS and heart rate monitors were used to reveal that warm-ups contribute significantly to the overall physical and physiological demands during matches [8]. Another wearable option is local positioning system (LPS), rather than GPS [30], to measure activity profiles. LPS works similarly to GPS, except that base stations placed around the field hockey pitch are used instead of satellites. Players are then required to wear active transponders using chest straps so that signals can be transmitted to the base stations.

Interestingly, the study that is most distinct from the others is one by Malan et al. [32], who examined thermoregulation responses in elite field hockey goalkeepers. These goalkeepers were required to wear a CorTemp data logger at the back of their pelvic guards after ingesting a radio-telemetry pill. This is also the only study to examine goalkeepers rather than outfield players, indicating that there is a lack of research into goalkeeper performances during competitions. Goalkeepers are generally excluded from activity profile analyses as their running distances are considered too short to determine performance [36]. However, as they are the least likely to be substituted throughout the game, it is essential that their performances are monitored during competitions to prevent any risk of injury, as well as to ensure that appropriate training plans can be provided for them. For instance, in Malan et al. [32], core temperatures of the goalkeepers increased before and after games in mild environment conditions. As goalkeepers are required to wear goalkeeping kit for protection, information pertaining to their core temperatures can be useful to coaches for ensuring that goalkeepers are properly hydrated, especially in hot and humid environments.

While GPS and heart rate monitors are the two most used devices in assessing player and team performance in field hockey matches, other measurements such as core temperature, hydration status, and injury risk are also of interest. Engineers and researchers should continue to develop innovative wearables to provide holistic information regarding players’ overall performance, well-being, and safety.

### 4.3. Performance Activity Bands

In the studies that analyzed player activity profiles, distance was the most commonly measured parameter as it was recorded by all studies using GPS technology. This was followed by speed, which was split into speed of locomotor activities (i.e., walking, running, sprinting) and speed of running intensity (i.e., low, moderate, high). Although the measured variables are similar, the different ranges of performance activity bands make comparison among studies rather difficult. Inconsistencies are immediately apparent when looking at the terms used to distinguish the categories. For example, in Macutkiewicz and Sunderland [38], locomotor activities are categorized into ‘standing’, ‘walking’, ‘jogging’, ‘running’, ‘fast running’, and ‘sprinting’. However, in the study by Morencos et al. [3], the first activity band was ‘standing-running’ and running activities were separated into ‘moderate speed running’ and ‘high speed running’. Others set the first activity band as ‘jogging’ and included ‘low speed running’, ‘moderate speed running’, and ‘sprinting’ [62]. This is also demonstrated in velocity bands, which were either categorized into levels (low, moderate, high) or velocity zones (Zone 1, Zone 2…, etc.) (Table 2). The inconsistency in these categorical terms means that some studies have more categories than others and could result in an overlap during comparison. For instance, if a study only has one ‘running’ category, it is unclear if this is comparable to moderate or fast running. Additionally, ‘jogging’ has a range of 6–11 km·h^−1^ in two studies [38,62], but 9.1–15.0 km·h^−1^ in another study [3]. Therefore, direct comparison between studies could be confusing, particularly if studies are assessed based on their categorical terms, and should be carried out with caution.

There is also a lack of a standardized range in the locomotor and velocity bands. The maximum threshold for standing in the study by Macukiewicz and Sunderland [38] was 6.0 km·h^−1^, while the threshold for Morencos et al. [3] was at 9.0 km·h^−1^. The locomotor activity between 15–19 km·h^−1^ that was categorized under ‘fast running’ in Macutkiewicz and Sunderland [38] was categorized as ‘moderate speed running’ in other studies [3,62]. ‘Jogging’ has a range of 6–11 km·h^−1^ in two studies [38,62] but 9.1–15.0 km·h^−1^ in another study [3]. This confusion was also seen in velocity bands, where moderate velocity in some studies [54,63] would be considered high velocity in another [38]. Speeds over 19 km·h^−1^ were considered sprinting in some studies [38], but others used higher cut-off speeds of 23 km·h^−1^ [3,54] or >30 km·h^−1^ [63]. Thus, a speed that is considered sprinting in the study of Romero-Moraleda et al. [54] would be considered as very high intensity running in the study by Casamichana et al. [63]. Owing to the inconsistent definitions among studies, results obtained from one study cannot be directly applied to others. Each study should be viewed individually with reference to how its performance bands were defined and categorized. Such information can be useful to coaches and sport scientists for implementing training programs that are relevant to their teams. 

Varied and conflicting activity bands are also seen within other variables such as acceleration and deceleration efforts, as well as heart rate bands. This suggests that, collectively, there is currently no consensus in performance activity bands for wearable devices in field hockey. Having standardized bands, however, may not be useful for match analyses as players of different teams might have different levels of physical fitness and speed thresholds. If standardized bands with fixed cut-offs are applied across different teams, the limit of each band could be too high for some players and too low for others. For instance, if a pre-determined moderate running intensity is set between 8.1–16.0 km·h^−1^, a player with higher aerobic capacity might consider running at 9.0 km·h^−1^ to be of low intensity. Nevertheless, the standardized band means that the player would be considered as running at moderate intensity during analysis. Thus, not taking relative aerobic capacity into consideration can result in inaccuracies during individual player analyses [63]. While generic thresholds are useful for providing information regarding a team’s ability to perform in competitions, future studies should develop new methods for setting performance bands using individual speed thresholds in order to better assess the relative demands of individual players.

### 4.4. Limitations

There are a few limitations to this study. Firstly, only wearables that are used in competitions are included in this review, thus eliminating any wearables that might have been used during training or in a less intense environment such as small-sided or simulated games. Moreover, during the screening process, it was discovered that many of the earlier papers were not available as full-text articles online and these studies were, therefore, excluded. Despite these limitations, the 39 studies in this review provided information regarding the competitive levels of the players that are most likely to use these wearables, as well as the types and purpose of wearables used during field hockey competitions.

### 4.5. Future Directions

This review provides an overview of the types of wearable devices used in field hockey competitions and the different parameters that these wearables are used to measure. Based on the results of the review, it is clear that there are inconsistencies within the devices (i.e., sampling rates) and within the parameters (i.e., difference in categories of the parameters) relevant to field hockey. Future studies should aim to use standardized wearable devices that can provide accurate, objective data for analysis. With a consensus that higher sampling rates provide higher accuracy, future studies should use GPS units of at least 10 Hz for more accurate analyses. Furthermore, individualized thresholds should be used to set performance bands, rather than simply adopting these bands from other studies, because players from different teams have different thresholds and capacities. Adopting performance bands from other studies assumes that all the players involved are similar in terms of their physical and physiological state. Researchers and practitioners could consider using peak speed as the limit of the highest performance band and adjust the other bands accordingly. It is also clear that while GPS and heart rate monitors are the most common wearable devices in field hockey, other wearable devices such as the CorTemp data logger could facilitate other types of analyses. The CorTemp data logger could also be used on outfield players to provide information regarding body temperature and how it could be better regulated. In addition, LPS provides indoor player movement tracking which could be used for indoor training, in the event that outdoor trainings are not available due to environmental constraints.

Furthermore, this review demonstrated that wearable technologies in field hockey are mostly used on elite athletes (Table 1). It is essential that elite athletes and teams perform at their best, especially in international competitions, and, therefore, use of wearable technologies to augment performance analysis is not uncommon. Future studies should also focus on non-elite athletes because improving their physical and physiological performances could allow them to contribute on a higher level. Lastly, wearable devices should be used to analyze goalkeeping performances. Goalkeepers are very different from outfield players in that they are required to wear extra goalkeeping kit, they are at risk of injury from fast incoming balls, and they are more likely to play the full game without substitution. Hence, more focus should be placed on their physical and physiological fitness to ensure that they are able to endure the demands of the competition.

## 5. Conclusions

Wearable technologies are currently being used in field hockey match analyses, with GPS and heart rate monitors being the most popular. Most studies examined player profile activities of field hockey players during international competitions at the elite level, but a lack of standardized performance bands makes comparisons between studies difficult. Furthermore, almost half of the reviewed studies were conducted using GPS units of low sampling rates, thus casting uncertainty on the accuracy of the results. With evidence showing higher accuracy in GPS units at higher sampling rates, future studies should consider a sampling rate of at least 10 Hz to provide sufficiently accurate data pertaining to player performance. Wearables should also be used more frequently at the non-elite level so that suitable training programs can be implemented to improve performance and to reduce the risk of injury. It has been observed that there is very little information on goalkeepers, who are excluded in all but one of the reviewed studies. Although goalkeepers are a small group compared to the outfield players, more emphasis should be placed on them as they also play an important role in the game. Wearable technologies are useful tools that can help coaches and sport scientists understand performance and adjust game strategies and training plans accordingly. Engineers and researchers should continue to develop innovative wearables to provide holistic information regarding players’ overall performance, well-being, and safety.

## Figures and Tables

**Figure 1 sensors-21-05242-f001:**
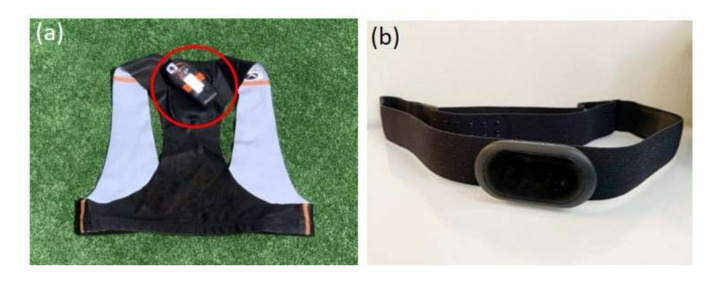
An example of (**a**) a global positioning system (GPS) unit (circled) with a vest, and (**b**) a heart rate monitor with chest strap.

**Figure 2 sensors-21-05242-f002:**
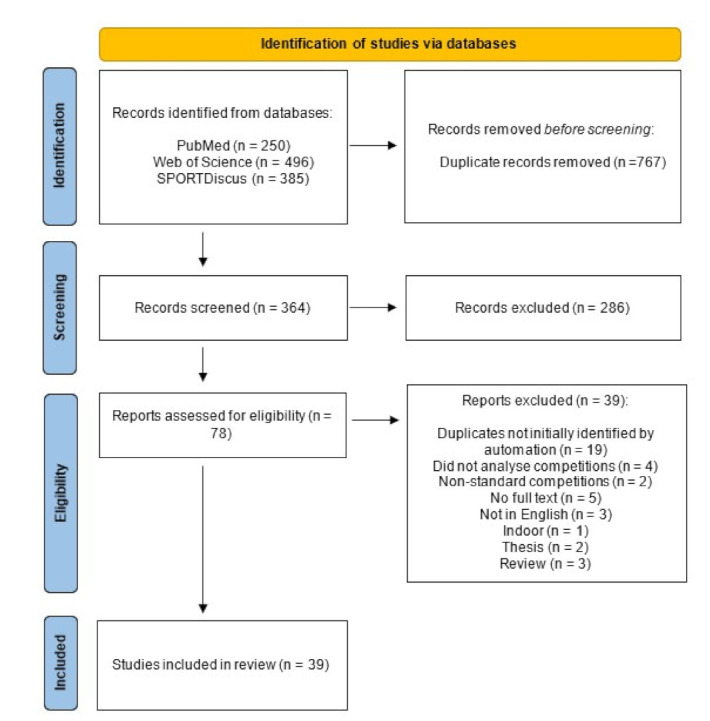
PRISMA flowchart.

**Figure 3 sensors-21-05242-f003:**
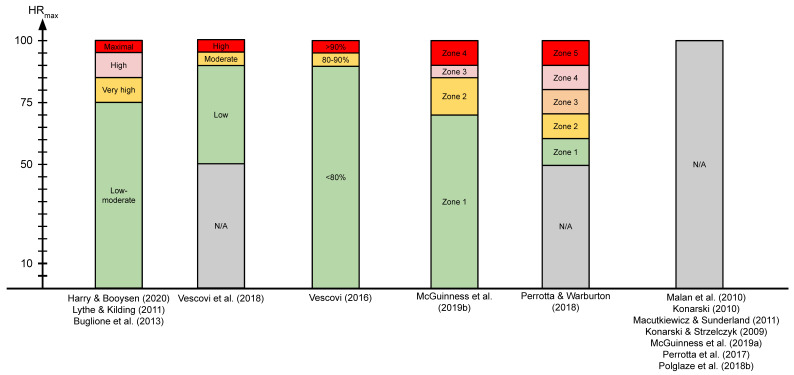
Heart rate bands in studies that used heart rate monitors. HR_max_: maximal heart rate.

**Table 1 sensors-21-05242-t001:** Study characteristics, wearable specifications, and measured parameters in field hockey competition analyses.

Wearable	Brand	Model	Sampling Rate	Competitive Level/Category/ Game Format	Study (Year) (Female/Male)	Dist	Time	Spd	Acc	Dec	Load	Met. Pwr	HR	Others
Active transponder	Inmotio	NA	1000	Elite/Open/2	Linke & Lames (2016) (M)	x	x							
GPS	Catapult	MiniMaxX	5	Elite/Open/2	Jennings et al. (2012a) (M)	x		x						
				Elite/Open/NA	Gabbett (2010) (M)	x		x	x					
				Elite/Open/2	Jennings et al. (2012b) (F)	x		x						
				Elite/Open/4	Ihsan et al. (2017) (M)	x		x	x	x				
				Elite/Open/2	White & MacFarlane (2013) (M)	x		x	x		x			
			10	Elite/Open/2	Polglaze et al. (2015) (M)	x			x		x			
				Elite/Open/4	Polglaze et al. (2018a) (M)	x		x	x			x		
				Elite/Open/4	Chesher et al. (2019) (M)	x						x		Deceleration bandsTime on pitchNo. of stints
				Elite/Open/2	Polglaze et al. (2018b) (M)	x	x					x		No. of stints
				Elite/Open/2	Warman et al. (2019) (M)		x							Total playing timePlaying time spent in each torso angle
		MiniMax Team 2.5	5	Elite/Open/4	Ihsan et al. (2021) (M)	x		x	x	x				
		OptimEye S5	10	Elite/Open/2	Warman et al. (2019) (M)		x							Total playing time Playing time spent in each torso angle
				Elite/Open/4	McGuinness et al. (2020) (F)	x	x	x						
				Elite/Open/4	McMahon & Kennedy (2019) (F)	x	x				x			No. of substitutions
	GPSports	SPI Elite	1	Elite/Open/2	Sunderland & Edwards (2017) (M)	x	x							No. of sprints
				Elite & Sub-elite/Open/2	Buglione et al. (2013) (M)	x		x	x	x				
				Elite/Open/2	Lythe & Kilding (2011) (M)	x		x						
			10	Sub-Elite/Open/4	Romero-Moraleda et al. (2020) (M)	x	x	x						
				Elite/Open/4	Morencos et al. (2019) (F)	x		x						
				Sub-Elite/Open/4	Morencos et al. (2018) (M)	x	x	x						
				Sub-Elite/Open/4	Casamichana et al. (2018) (M)	x	x	x						
			NA	Elite/Open/2	Macutkiewicz & Sunderland (2011) (F)	x	x	x						
		SPI HPU	5	Elite/Open/NA	Kim et al. (2018) (F)	x		x						
		SPI Pro	5	Elite/Youth/2	Vescovi (2014) (F)			x						
				Sub-Elite/Youth/2	Vescovi & Frayne (2015) (F)	x	x	x	x	x				Work rate
				Elite/Youth/2	Vescovi & Klas (2018) (F)	x								
				Elite/Youth/2	Vescovi (2016) (F)	x		x				x		
		SPI ProX	15	Sub-Elite/Open/NA	Wergin et al. (2020) (M)	x								
		SPI HPU	15	Novice to Elite/Open 2	Vinson et al. (2018) (F)	x	x	x						
	VXSports	NA	4	Elite/Open/2	McGuinness et al. (2019b) (F)	x								
		VX 110	10	Elite/Open/4	McGuinness et al. (2019a) (F)	x		x						Workload
		VX 350b	10	Sub-elite/Youth/4	van der Merwe & Haggie (2019) (M)	x		x						
	STATSports	Apex	10	Elite/Open/4	McGuinness et al. (2021) (F)	x		x	x	x				
HR Monitor	Polar	NA	NA	Elite & Sub-elite/Open/2	Buglione et al. (2013) (M)								x	
		Team	NA	Elite/Open/2	Lythe & Kilding (2011) (M)								x	
		T31	5000	Elite/Youth/2	Vescovi et al. (2018) (F)								x	
		Team 2	NA	Elite/Open/2	McGuinness et al. (2019b) (F)								x	
		T31	NA	Elite/Open/4	McGuinness et al. (2021) (F)								x	
		Team 2	1000	Elite/Open/2	Perrotta & Warburton (2018) (F)		x						x	
		Team 2	1000	Elite/Open/2	Perrotta et al. (2017) (F)						x		x	
		Vantage	NA	Elite/Open/NA	Konarski (2010) (NA)								x	EEE
		Team	NA	NA/Open/2	Konarski & Strzelczyk (2009) (M)								x	
		NA	NA	Elite/Open/2	Malan et al. (2010) (F&M)								x	
		Team 2	NA	Elite/Open/2	Polglaze et al. (2018b) (M)								x	
	Firstbeat Technologies Oy	NA	NA	Elite/Open/4	McGuinness et al. (2019a) (F)								x	
	Zephyr	BioHarness 3	250	Sub-elite/Open/2	Harry & Booysen (2020) (F)								x	
CorTemp Logger	Not Mentioned	NA	NA	Elite/Open/2	Malan et al. (2010) (F&M)									Core and skin temperature
Accelerometer	Zephyr	BioHarness 3	250	Sub-elite/Open/2	Harry & Booysen (2020) (F)		x							

Game formats include either two halves (35 min each) or four quarters (15 min each); GPS: Global Positioning System; F: Female; M: Male; Dist: Distance; Spd: Speed; Acc: Acceleration; Dec: Deceleration; Met. Pwr: Metabolic Power; HR: Heart rate; EEE: Estimated Energy Expenditure.

**Table 2 sensors-21-05242-t002:** Performance activity bands in studies that used global positioning system (GPS), active transponder devices, or Zephyr BioHarness.

Study	Locomotor Bands	Velocity Bands	Estimated Energy	Acceleration/Deceleration Efforts
Linke & Lames (2016)	N/A	N/A	N/A	N/A
Jennings et al. (2012a)	N/A	LSA: 0.10–4.17 m·s^−1^ HSR: >4.17 m·s^−1^	N/A	N/A
Gabbett (2010)	N/A	Low: 0–1 m·s^−1^ Moderate: 1–5 m·s^−1^ High: >5 m·s^−1^	N/A	N/A
Jennings et al. (2012b)	N/A	LSA: 0.10–4.17 m·s^−1^ HSR: >4.17 m·s^−1^	N/A	N/A
Ihsan et al. (2017)	N/A	N/A	N/A	N/A
White & MacFarlane (2013)	DNM: 0–6 km·h^−1^ Jogging: 6–11 km·h^−1^ LSR: 11–15 km·h^−1^ MSR: 15–19 km·h^−1^ DNM: 19–23 km·h^−1^ Sprinting: >23 km·h^−1^	N/A	N/A	High acceleration: >2.0 m·s^−2^
Ihsan et al. (2021)	N/A	N/A	N/A	N/A
Polglaze et al. (2015)	N/A	N/A	N/A	N/A
Polglaze et al. (2018a)	N/A	Low speed: <15.5 km·h^−1^ High speed: >15.5 km·h^−1^	High power: >20.0 W·kg^−1^ Low power: <20.0 W·kg^−1^	High acceleration: >2.0 m·s^−2^ Low acceleration: <2.0 m·s^−2^
Chesher et al. (2019)	N/A	N/A	N/A	Band 1 (Low): −3 to −5.99 m·s^−2^ Band 2 (Medium): −6 to −8.99 m·s^−2^ Band 3 (High): −9 to −11.99 m·s^−2^ Band 4 (Very high): <−12 m·s^−2^
Polglaze et al. (2018b)	N/A	N/A	N/A	N/A
Warman et al. (2019)	N/A	N/A	N/A	N/A
McGuinness et al. (2020)	N/A	Zone 1: <7.9 km·h^−1^ Zone 2: 8–10.9 km·h^−1^ Zone 3: 11–15.9 km·h^−1^ Zone 4: >16 km·h^−1^ Zone 5: >20 km·h^−1^	N/A	N/A
McMahon & Kennedy (2019)	N/A	LSR: 0–3.05 m·s^−1^ HSR: 3.08–5.27 m·s^−1^	N/A	N/A
Sunderland & Edwards (2017)	N/A	N/A	N/A	N/A
Buglione et al. (2013)	N/A	Zone 1:0.1–6 km·h^−1^ Zone 2: 6.1–11 km·h^−1^ Zone 3: 11.1–14 km·h^−1^ Zone 4: 14.1–19 km·h^−1^ Zone 5: 19.1–23 km·h^−1^ Zone 6: >23 km·h^−1^	N/A	1 < a < 2 m·s^−2^ 2 < a < 3 m·s^−2^ >3 m·s^−2^ −1 < a < −2 m·s^−2^ −2 < a < −3 m·s^−2^ <−3 m·s^−2^
Lythe & Kilding (2011)	N/A	Zone 1: 0–6 km·h^−1^ Zone 2: 6.1–11 km·h^−1^ Zone 3: 11.1–14 km·h^−1^ Zone 4: 14.1–19 km·h^−1^ Zone 5: 19.1–23 km·h^−1^ Zone 6: >23 km·h^−1^	N/A	N/A
Kim et al. (2018)	N/A	Very low: <6 km·h^−1^ Low: 6–12 km·h^−1^ Moderate: 12.1–18 km·h^−1^ High: 18.1–24 km·h^−1^ Very high: >24 km·h^−1^	N/A	N/A
Vescovi (2014)	N/A	Slower players: ≤29.0 km·h^−1^ Faster players: ≥29.0 km·h^−1^	N/A	N/A
Vescovi & Frayne (2015)	N/A	Low: 0–8.0 km·h^−1^ Moderate: 8.1–16.0 km·h^−1^ High: 16.1–20.0 km·h^−1^ Maximal: 20.1–32.0 km·h^−1^	Low: ≤10 W·kg^−1^ Intermediate: 10.1–20 W·kg^−1^ High: 20.1–35 W·kg^−1^ Elevated 35.1–55 W·kg^−1^ Maximal: >55 W·kg^−1^	N/A
Vescovi et al. (2018)	N/A	Low: 0–8.0 km·h^−1^ Moderate: 8.1–16.0 km·h^−1^ High: 16.1–20.0 km·h^−1^ Maximal: >20.0 km·h^−1^	N/A	N/A
Vescovi (2016)	N/A	Low: 0–8.0 km·h^−1^ Moderate: 8.1–16.0 km·h^−1^ High: 16.1–20.0 km·h^−1^ Maximal: >20.1–32.0 km·h^−1^	Low: ≤10 W·kg^−1^ Intermediate: 10.1–20 W·kg^−1^ High: 20.1–35 W·kg^−1^ Elevated 35.1–55 W·kg^−1^ Maximal: >55 W·kg^−1^	N/A
Romero-Moraleda et al. (2020)	N/A	LSR: <15.0 km·h^−1^ MSR: 15.1–18.9 km·h^−1^ HSR: >19 km·h^−1^ SR: >23.0 km·h^−1^	N/A	Low: 1–1.9 m·s^−2^ Moderate: 2–2.9 m·s^−2^ High: >3 m·s^−2^
Morencos et al. (2019)	N/A	N/A	N/A	Low: 1–1.9 m·s^−2^ Moderate: 2–2.9 m·s^−2^ High: >3 m·s^−2^
Morencos et al. (2018)	Standing-Walking: <9.0 km·h^−1^ Jogging: 9.1–15.0 km·h^−1^ MSR: 15.1–19 km·h^−1^ HSR: >19 km·h^−1^ Sprinting: >23 km·h^−1^	N/A	N/A	Low: 1–1.9 m·s^−2^ Moderate: 2–2.9 m·s^−2^ High: >3 m·s^−2^
Casamichana et al. (2018)	N/A	MSR: 15.1–18.9 km·h^−1^ HSR: 19–23.9 km·h^−1^ VHSR: 24–29.9 km·h^−1^ SR: >30 km·h^−1^	N/A	N/A
Macutkiewicz & Sunderland (2011)	Standing: 0–0.6 km·h^−1^ Walking: 0.7–6.0 km·h^−1^ Jogging: 6.1–11.0 km·h^−1^ Running: 11.1–15.0 km·h^−1^ Fast running: 15.1–19.0 km·h^−1^ Sprinting: >19.0 km·h^−1^	Low: 0–6 km·h^−1^ Moderate: 6.1–15.0 km·h^−1^ High: 15.1–29.5 km·h^−1^	N/A	N/A
McGuinness et al. (2019b)	N/A	Zone 1: 0–7.9 km·h^−1^ Zone 2: 8–15.9 km·h^−1^ Zone 3: 16–19.9 km·h^−1^ Zone 4: >20 km·h^−1^	N/A	N/A
Wergin et al. (2020)	N/A	N/A	N/A	N/A
Vinson et al. (2018)	N/A	Zone 1: 0% MSS Zone 2: 0.1–20% MSS Zone 3: 20.1–35% MSS Zone 4: 35.1–50% MSS Zone 5: 50.1–70% MSS Zone 6: >70% MSS	N/A	N/A
McGuinness et al. (2019a)	N/A	Zone 1: <7.9 km·h^−1^ Zone 2: 8–10.9 km·h^−1^ Zone 3: 11–15.9 km·h^−1^ Zone 4: 16–19.9 km·h^−1^ Zone 5: >20 km·h^−1^	Low: >7.9 km·h^−1^ Moderate: 8–15.9 km·h^−1^ High: >16 km·h^−1^	N/A
van der Merwe & Haggie (2019)	N/A	Band 1: ≤5.9 km·h^−1^ Band 2: 6–10 km·h^−1^ Band 3: 10.1–14.9 km·h^−1^ Band 4: 15–24.6 km·h^−1^ Band 5: ≥24.7 km·h^−1^	N/A	N/A
McGuinness et al. (2021)	N/A	Zone 1: 0–7.9 km·h^−1^ Zone 2: 8–15.9 km·h^−1^ Zone 3: 16–19.9 km·h^−1^ Zone 4: >20 km·h^−1^	N/A	N/A
Harry & Booysen (2020)	Standing and walking: <0.8 VMU Jogging: 0.8–1.09 VMU Running: 1.1–1.39 VMU Sprinting: ≥ 1.4 VMU	N/A	N/A	N/A

DNM: did not mention band name; LSA: low speed activity; LSR: low speed running; MSR: moderate speed running; HSR: high speed running; VHSR: very high speed running; SR: sprinting; a: acceleration; MSS: maximal sprint speed; VMU: vector magnitude units measured by triaxial accelerometer in the Zephyr BioHarness.

## Data Availability

No new data were created or analyzed in this study. Data sharing is not applicable to this article.

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
