# Peer review of "Wearable Technologies in Field Hockey Competitions: A Scoping Review"

_sensors, 2021, doi:10.3390/s21155242_

Round 1

Reviewer 1 Report

The introduction section is weak and not persuasive. More specifically, why wearable technology is important for field hockey. Are there some specific reasons for using wearable technologies for athletes?

Moreover, the research gap is not clear.  Although the authors mentioned that "no study has collectively examined all wearable technologies that field hockey players might wear during their competitions. ", the justification is not strong. Maybe the authors can address more the potential theoretical and practical implications to emphasize the importance of this study. 

The Literature review is missing. How wearable technologies have been applied and used in sport? The application of wearable technologies in sport has been studied extensively in recent years. The authors need to review the relevant studies and address the gap need for this study. 

The methodology and results are well-reported and interpreted. 

The discussion is well-written with much in-depth information. However, it would be better to address some theoretical implications of this study and suggestions for future applications of wearable technology in sport. 

Author Response

Thank you for reviewing our paper. Please see our point-by-point reply in the attached document.

Reviewer 2 Report

sensors-1266749

Recommendation: Accept with Major Revision

Rationale: This is the Journal of Sensors at MDPI. Yet, there is no information, at least readily seen,  on or about the composition of the GPS sensors. There is no information on circuitry either. There is no information on how it works. Granted the paper gives an excellent review on information about sensors used in field hockey but the Journal of Sensors at MDPI provides information about sensors. Should the authors provide the necessary information with proper citations and credits, the paper will be reconsidered for publication in the prestigious Journal of Sensors.  

Thank you for your interest in Sensors.

Author Response

(The authors gave the same response as above.)

Reviewer 3 Report

This paper presented the review of wearables currently used during field hockey competitions. The topic of this paper fits the scope of the journal, and the review results are promising to the study of wearable sensing technology in physical and physiological performance measurement during high-intensity exercise. However, the current quality of this paper is too low. Therefore, a major revision is required before the paper can be accepted for publication.

  • The abstract needs to be revised to more fit the scope of the journal and address the importance of the research.
  • Although the manuscript provides some simple quantitative comparisons, these comparisons are limited to some apparent content discussions. Therefore, it needs to have an in-depth and thorough discussion on the current sensor applications and the sensing advantages and difficulties faced by them, including but not limited to sensing accuracy, efficiency, ease of use, etc.
  • The author needs to conduct an in-depth discussion on the current field of sensing technology and give feasible development strategies and technical directions so as to give readers in-depth research and application guidance.
  • The current manuscript needs to discuss the sensing qualities of different sensor types and sensor signals, as well as the sensor problems that are easily caused by the current application scenarios.
  • The author needs to give some physical pictures and practical application scene diagrams to facilitate readers to obtain detailed sensing details such as sensors' appearance and their wearing methods.
  • Some tables should be replaced by histograms or scatter plots to achieve a more intuitive and effective description of the research status.
  • The manuscript needs to add a new chapter to describe the logical connection between the current sports scene and the sensor type or the sensing technology used.

Author Response

(The authors gave the same response as above.)

Round 2

Reviewer 1 Report

In general, I am satisfied with the revised version of the manuscript. However, the search strategy is not clear. The authors mentioned that "The search strategy combined keywords that included field hockey, wearables, accelerometers, inertial sensors, GPS, heart rate monitors, load, performance analysis, player activity profiles, and competitions.

It is not clear how the authors generate the results. Which field did you search? (abstract, title, keywords?)  What was your search string? Did you use  "OR" "AND" in your search? 

Please further clarify this in your further revision. 

Author Response

The authors thank the reviewer for the positive feedback. We used [All fields] in the search. The search strategy has been extended under Section 2.1. to include the combination of keywords (Lines 163-170).

“The search strategy included a combination of keywords [All Fields]: ‘field hockey’ AND (wearable* OR technolog*) OR (acceleromet*) OR (GPS OR global positioning system* OR global positioning satellite* OR global positioning unit* OR global positioning device*) OR (inertial senor* OR inertial measurement) OR (‘heart rate’ monitor OR HR monitor) OR ‘load’ OR (performance AND analysis OR ‘game analysis’ OR ‘motion analysis’ OR ‘time motion analysis’) OR (movement OR ‘player activity’ OR profiles OR patterns) AND (match* OR competition* OR tournament*) OR (‘game play’ OR ‘match play’).”

Reviewer 2 Report

Recommendation: Accept in present form.

Comments: One of the best, detailed and accurate responses to reviewers' questions and comments that that this reviewer, author, writer, scientist, editor and Academic Editor has ever seen! Each and every response to constructive query is delineated in its finest form. 

Serious sensor scientists are evidenced for our prestigious Journal of Sensors.

The article begets interest in field hockey and soccer forms of athletics as these relate to the burgeoning and incredibly wonderful newness of the fascinating field of sensors.

Congratulations from one sensor person to another! Sensors make sense!

Author Response

The authors would like to thank the reviewer for the positive feedback. 

Reviewer 3 Report

The revision is fine.

Author Response

The authors would like to thank the reviewer for accepting our revisions.